# IMPLICIT IN-CONTEXT LEARNING

**Zhuowei Li, Zihao Xu, Ligong Han, Yunhe Gao, Song Wen, Di Liu
Hao Wang, Dimitris N. Metaxas**
Department of Computer Science
Rutgers University
`{<first name>.<last name>}@rutgers.edu`

## ABSTRACT

In-context Learning (ICL) empowers large language models (LLMs) to swiftly adapt to unseen tasks at inference-time by prefixing a few demonstration examples before queries. Despite its versatility, ICL incurs substantial computational and memory overheads compared to zero-shot learning and is sensitive to the selection and order of demonstration examples. In this work, we introduce **Implicit In-context Learning** (I2CL), an innovative paradigm that reduces the inference cost of ICL to that of zero-shot learning with minimal information loss. I2CL operates by first generating a condensed vector representation, namely a context vector, extracted from the demonstration examples. It then conducts an inference-time intervention through injecting a linear combination of the context vector and query activations back into the model's residual streams. Empirical evaluation on nine real-world tasks across three model architectures demonstrates that I2CL achieves few-shot level performance at zero-shot inference cost, and it exhibits robustness against variations in demonstration examples. Furthermore, I2CL facilitates a novel representation of "task-ids", enhancing task similarity detection and fostering effective transfer learning. We also perform a comprehensive analysis and ablation study on I2CL, offering deeper insights into its internal mechanisms. Code is available at `https://github.com/LzVv123456/I2CL`.

## 1 INTRODUCTION

In-context Learning (ICL) has emerged as a prominent capability of large language models (LLMs) (Brown et al., 2020). It enables a swift inference-time adaptation towards new tasks by prefixing a few demonstration examples prior to the query (Wei et al., 2022; Dong et al., 2023). ICL, characterized by its adaptability, has prompted extensive research efforts aimed at optimizing the selection of demonstration examples, or prompts (Rubin et al., 2022; Sorensen et al., 2022; Wu et al., 2023; Min et al., 2022a, *inter alia*), as well as mitigating their sensitivity to formatting, order, and recency bias (Zhao et al., 2021; Lu et al., 2022; Hao et al., 2022, *inter alia*).

Despite existing advances, current approaches predominantly focus on manipulating demonstration examples within the token space, where context tokens are prepended to the query. This practice quadratically escalates the computation and memory demands with each additional token and is known to be sensitive to the selection and order of demonstration examples Zhao et al. (2021); Lu et al. (2022); Dong et al. (2023). Consequently, it is challenging to apply ICL under constrained scenarios where computational and memory resources are scarce, and demonstration profiles are uncontrollable, hindering ICL's scalability and practical utility.

In this study, we explore to harness the demonstration examples within the activation space, seeking for an efficient and robust alternative for constrained environments. Given a decoder-only architecture, we observe that the primary burdens of ICL arise from the computationally intensive multi-head attention mechanism, which fuses information between demonstration and query tokens, and the memory-intensive key-value caching scheme necessary for retaining contextual information[1]. These observations motivate us to investigate the following two questions: *Is there a more abstract repre-*

---

[1]Without applying key-value cache, one needs to repetitively forward the same demonstration examples.

*sentation of demonstration examples?* And, *can we integrate such information into models without resorting to attention mechanism?*

Our findings suggest that both objectives are attainable by condensing demonstration examples into a compact vector representation and reintegrating their functionalities within the model's activation space. Instead of concatenating demonstration tokens before the query tokens, we independently extract a *demonstration vector* from each example. These demonstration vectors are aggregated in a permutation-invariant manner to form a unified *context vector*. During inference, a linear combination of the context vector and the query activations is injected into the model's residual streams as the substitution of the original output activations. We term above scheme **Implicit In-context Learning** (I2CL), alluding to the absence of explicit demonstration tokens at querying stage.

I2CL offers unprecedented merits. By condensing demonstration examples into a unified vector representation, I2CL needs to cache only a fixed amount of activation vectors, independent to the number of demonstration tokens. I2CL maintains a zero-shot inference speed through merging information between demonstration examples and queries using only linear operators. We evaluate I2CL across three open-source LLMs on nine real-world text classification tasks, where it significantly outperforms zero-shot counterpart and other comparable methods. I2CL achieves results on par with few-shot learning with zero-shot inference cost (see Fig. 1). Importantly, I2CL demonstrates robustness against the variability of demonstration examples, and facilitates a natural representation of *task-ids* that can effectively signify task similarities and foster transfer learning.

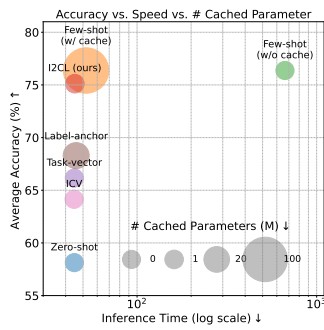

Figure 1: Upper-left is better. Comparison of accuracy, inference speed and cached memory of different methods on Llama2-7b.

Our main contributions can be summarized in three-fold: (1) We introduce I2CL, a novel framework that effectively integrates a minimal set of demonstration examples within the activation space. By decomposing conventional ICL into two stages: context vectorization and context-vector injection, I2CL attains few-shot level performances at zero-shot inference cost. (2) We empirically validate the robustness of I2CL against the variations (*i.e.*, choices and order) of demonstration examples and uncover a natural generation of task-ids, upon which we further propose a transfer learning strategy that can enhance performance on new tasks based on existing anchors. (3) We conduct a comprehensive analysis and ablation to thoroughly examine each component and design choice of I2CL, thereby shedding light on I2CL's internal working mechanisms.

## 2 METHODOLOGY

### 2.1 PRELIMINARIES

**Residual Stream** We adopt the mathematical interpretation from Elhage et al. (2021), viewing the hidden states across layers and at each token position as a residual stream. This perspective treats each attention head and multi-layer perceptron (MLP) module as read-out and write-in operators that engage with residual streams, facilitating the addition and deletion of information within residual streams. At a given layer $l$ and token position $t$, the residual stream $r_l^t$ is defined recursively as:

$$r_l^t = r_{l-1}^t + a_l^t + m_l^t, \quad (1)$$

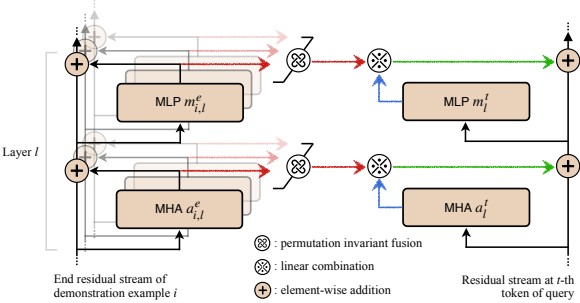

Figure 2: A schematic overview of I2CL, including a single layer for illustrative purpose.

where $a_l^t$ denotes the integrated output of the multi-head attention (MHA) module, and $m_l^t$ signifies the MLP's output, contingent on $a_l^t$. Within this framework, MHA promotes the information fusion across residual streams, whereas the MLP functions akin to an associative memory (Geva et al., 2021; Dai et al., 2022) that retrieves information encapsulated within its weights.

**In-context Learning**  Given an unseen task, ICL assumes the existence of a set of $N$ demonstration examples $\mathbb{D} = \{d_1, d_2, \ldots, d_N\}$, each comprising an instructional pair $d_i = (x_i, y_i)$ that includes an input sentence[2] and its corresponding label. ICL operates by first concatenating the demonstration set $\mathbb{D}$ with the test query $x_q$, forming an input prompt $p = [\mathbb{D}, x_q]$. The objective is then to predict the correct response $y_q$ from a finite set of discrete labels $\mathbb{C}$ via $y_q = \arg\max_{y \in \mathbb{C}} P(y \mid p)$. We acknowledge the versatility and broader implication of ICL, especially when connecting to the generic definition of prompt engineering. In this work, we consider the standard few-shot classification task as our testbed and focus on this scenario in the following sections.

## 2.2 Context Vectorization

To overcome the inefficiencies associated with key-value caching system, we isolate the reasoning process of demonstration examples from that of the query, and introduce an independent vectorization process for each demonstration pair $d_i$. Extracted vectors are merged in the activation space. Specifically, we define a function $V$, of generating a vector representation for each demonstration example: $\boldsymbol{d}_i = V(d_i)$, followed by a function $F$ applied to aggregate extracted demonstration vectors into a unified context vector $\boldsymbol{v} = F(\boldsymbol{d}_{i=1}^N)$. Note that $F$ is designed to be permutation invariant, ensuring a unique vector representation for a given set of demonstration examples.

In our implementation, function $V$ is parameterized by a pre-trained LLM and its corresponding tokenizer, and operates by collecting output activations of both MHA and MLP modules across layers at the position of end residual stream:

$$\boldsymbol{d}_i = \{\boldsymbol{a}_{i,l}^e, \boldsymbol{m}_{i,l}^e\}_{l=1}^L \,{}^3. \tag{2}$$

Here, $e$ denotes the last token position that is responsible for next-token prediction, and $L$ represents the total number of layers. For the aggregation function, we compute an element-wise arithmetic mean of each vector component across the demonstration examples:

$$\boldsymbol{v} = \{\bar{\boldsymbol{a}}_l^e, \bar{\boldsymbol{m}}_l^e\}_{l=1}^L \tag{3}$$

where $\bar{\boldsymbol{a}}_l^e = \frac{1}{N} \sum_{i=1}^N \boldsymbol{a}_{i,l}^e$ and $\bar{\boldsymbol{m}}_l^e = \frac{1}{N} \sum_{i=1}^N \boldsymbol{m}_{i,l}^e$.

The rationale of these designs is twofold. First, we argue that the end residual stream encapsulates the essential information from each example. This is supported by the dynamics of next-token prediction and empirical findings from recent studies (Hendel et al., 2023; Zou et al., 2023; Todd et al., 2024). Second, inspired by the linear representation hypothesis (Park et al., 2023), we premise various demonstration vectors can undergo linear transformations at different levels of abstraction.

## 2.3 Context Injection

Having established a unique vector representation, $\boldsymbol{v}$, for the demonstration set, I2CL seeks to enhance zero-shot performance by integrating the contextual information. While conventional methods leverage the multi-head attention module for this purpose, I2CL utilizes a simpler, yet effective, linear operation to augment the query activations with context vectors.

Taking residual stream $\boldsymbol{r}_l^t$ as an instance, instead of directly adding the output activations from the MHA and MLP to $\boldsymbol{r}_l^t$, we inject a linear combination of these activations with context vectors:

$$\boldsymbol{r}_l^t = \boldsymbol{r}_{l-1}^t + (\lambda_l^a \bar{\boldsymbol{a}}_l^e + \beta_l^a \boldsymbol{a}_l^t) + (\lambda_l^m \bar{\boldsymbol{m}}_l^e + \beta_l^m \boldsymbol{m}_l^t), \quad l \in [1, L], t \in [1, T]. \tag{4}$$

Here, $\lambda^a, \beta^a, \lambda^m, \beta^m$ are four layer-wise scalars used to adjust the proportion to which the context vectors and query activations are blended. By default, we apply this information fusion process to all residual streams of a given query. Letting $\lambda = 0$ and $\beta = 1$ (omitting subscripts) replicates the original zero-shot inference process. Note that it is critical to extract and aggregate information at both MHA and MLP modules (see Section 3.4), suggesting a distinct yet complementary functionality of MHA and MLP.

The proposed context injection method incurs minimal computational overhead, involving only two scalar multiplications and an element-wise addition, a process more efficient than the attention mechanism. As a result, the inference speed of I2CL matches that of standard zero-shot learning. We refer to Figure 2 for a schematic illustration of the context vectorization and injection procedures.

---

[2]Input $x_i$ also includes formatting texts like: "Question:", "Answer Type:".

[3]We abuse the symbolic notion of a vector to denote a set for the ease of interpretation.

## 2.4 NOISY SELF-CALIBRATION

Thus far, we have presented how to vectorize and reintegrate the contextual information in an efficient and attention-free manner. Herein, we elaborate on how to configure the scale of linear coefficients. It is not uncommon to set those weight scalars as hyper-parameters (Turner et al., 2023; Liu et al., 2024), and manually adjust them for each task in a trial and error fashion. In order to achieve an adaptive and nuanced control over the information fusion process without human-in-the-loop, we propose estimating the linear coefficients $\boldsymbol{c} = \{\lambda_l^a, \beta_l^a, \lambda_l^m, \beta_l^m\}_{l=1}^L$ using a gradient-based optimization applied to the same set of demonstration examples. Our estimation of linear coefficients does not require any additional data beyond the given few-shot examples.

Concretely, we initialize $\lambda = 0.1, \beta = 1.0$ to promote a modest initial addition of information, and update these coefficients by minimizing the perplexity of label tokens:

$$\mathcal{L} = -\frac{1}{|\mathbb{D}|} \sum_{(x,y) \in \mathbb{D}} \log P(y \mid x, \boldsymbol{v}, \boldsymbol{c}), \tag{5}$$

where $P(\cdot)$ denotes the induced probability distribution over the entire vocabulary at the end token position from the last layer. To bolster the robustness and adaptability of estimated linear coefficients to potential downstream variations, we perturb the residual streams with Gaussian noises $\boldsymbol{\eta} \sim \mathcal{N}(\boldsymbol{0}, \boldsymbol{I})$ during the calibration phase:

$$\begin{aligned}
\boldsymbol{o}_l^t &= \boldsymbol{r}_{l-1}^t + (\lambda_l^a \bar{\boldsymbol{a}}_l^e + \beta_l^a \boldsymbol{a}_l^t), \\
\boldsymbol{o}_l^t &= \boldsymbol{o}_l^t + \gamma ||\boldsymbol{o}_l^t||_2 \times \boldsymbol{\eta}, \\
\boldsymbol{r}_l^t &= \boldsymbol{o}_l^t + (\lambda_l^m \bar{\boldsymbol{m}}_l^e + \beta_l^m \boldsymbol{m}_l^t), \\
\boldsymbol{r}_l^t &= \boldsymbol{r}_l^t + \gamma ||\boldsymbol{r}_l^t||_2 \times \boldsymbol{\eta},
\end{aligned} \tag{6}$$

where $\gamma$ is a scalar employed to modulate the intensity of the noise, and $|| \cdot ||_2$ denotes the $L^2$ norm. The $\boldsymbol{o}$ represents the intermediate state of a residual stream.

Given above formulations, only a few linear coefficients (totaling $4L$) are updated during the calibration phase, rendering this process remarkably efficient (consuming 1-2 minutes on a single A100 40G). Critically, the calibrated linear coefficients are demonstration agnostic—they require calibration only once per task and exhibit excellent generalization ability to unseen demonstration examples (see Section 3.3). Moreover, these linear coefficients, though are lightweight, can function as effective task-ids that fostering task similarity detection (Fig. 5) and transfer learning (Table 3).

## 3 EXPERIMENTS

In this empirical section, we begin by detailing the architectures, tasks, and configurations used in our study, followed by a comparative analysis of I2CL against other relevant techniques. We then delve into the formation of context vectors and the characteristics of the calibrated linear coefficients, demonstrating the robustness of I2CL, as well as identifying the function of calibrated coefficients as task-ids. This section concludes with an extensive ablation study to underscore the inner working mechanism of I2CL. We refer readers to Appendix C for additional experiments and analysis.

**Models** We evaluate I2CL on three open-source architectures: GPT2-XL (Radford et al., 2019), GPT-J-6B (Wang & Komatsuzaki, 2021), and Llama2-7b (Touvron et al., 2023). We selected these models based on their suitability for our computational resources and their range in size from relatively small (1.5B) to large (7B). We report results under Llama2-7b in the main content and defer results of other architectures to Appendix C.1. Consistent trends are observed across all architectures.

**Tasks** We first take the four tasks used in Wang et al. (2023), including sentiment analysis: SST-2 (Socher et al., 2013), emotion classification: EmoC (Chatterjee et al., 2019), question classification: TREC (Voorhees & Tice, 2000), and topic classification: AGNews (Zhang et al., 2015). We then enrich our experiments with five additional datasets, encompassing 5-way sentiment analysis: SST-5 (Socher et al., 2013), movie review classification: MR (Pang & Lee, 2005), 14-way topic classification: DBPedia (Zhang et al., 2015), subjectivity status categorization: Subj (Pang & Lee, 2004), and hate speech detection: hate_speech18 (de Gibert et al., 2018). We employ the HuggingFace version of the data (Lhoest et al., 2021) and sample 500 data points from the validation/test set for evaluation.

Table 1: Comparison between I2CL and baseline methods on Llama2-7b. The **best** results are highlighted in bold, and the second-best results are underlined. In addition to a practical gauge of the inference speed and memory usage (see Fig 1), we include an examination of cached parameters. Here, $M$, $D$, and $L$ denote the number of demonstration tokens, model dimension, and architecture layers, respectively. $P$ indicates the number of extra learnable tokens in the Soft-prompt method, and $1/K$ represents the compression rate of corresponding context-compression method.

| Method | SST-2 (%) ↑ | SST-5 (%) ↑ | TREC (%) ↑ | AGNews (%) ↑ | Subj (%) ↑ | HateSpeech18 (%) ↑ | DBPedia (%) ↑ | EmoC (%) ↑ | MR (%) ↑ | Avg. acc. (%) ↑ | # cached param. ↓ |
|---|---|---|---|---|---|---|---|---|---|---|---|
| Zero-shot | 83.00 | 27.00 | 50.00 | 70.20 | 51.40 | 54.20 | 72.00 | 41.80 | 73.60 | 58.13 | 0 |
| Few-shot (ICL) | 94.44±1.44 | 41.72±3.68 | 77.32±4.41 | 85.68±2.00 | 52.56±3.09 | 70.24±5.80 | 96.64±0.48 | 75.48±1.63 | 93.24±0.50 | 76.37 | $2MDL$ |
| Noise vector | 49.88±0.24 | 20.56±0.64 | 20.12±10.92 | 27.32±2.82 | 49.64±0.48 | 59.84±8.04 | 7.28±0.37 | 26.76±3.04 | 50.12±0.24 | 34.61 | $2DL$ |
| Label-anchor | 83.32±5.95 | 27.68±4.21 | 77.48±3.49 | 83.72±1.04 | 53.00±2.95 | 64.52±8.09 | 81.40±3.67 | 59.12±10.60 | 84.40±5.89 | 68.29 | $2(M/K)DL$ |
| Task-vector | 81.44±4.73 | 25.96±0.59 | 65.68±1.93 | 79.68±4.07 | 58.56±4.91 | 67.68±3.70 | 89.48±2.58 | 44.64±3.53 | 82.32±5.37 | 66.16 | $D$ |
| ICV | 86.28±0.55 | 33.48±0.65 | 63.84±0.15 | 72.40±0.37 | 56.56±0.70 | 60.56±1.50 | 73.64±0.88 | 49.16±1.24 | 84.04±1.10 | 64.44 | $DL$ |
| **I2CL (ours)** | **87.68±2.47** | **39.12±2.69** | **78.56±5.32** | **85.48±1.16** | **73.84±3.84** | **69.88±5.67** | **90.16±1.86** | **63.72±1.37** | **87.68±2.26** | **75.12** | $2DL$ |
| AutoComp. | 92.44±3.29 | 25.8±4.8 | 62.52±9.34 | 86.36±1.03 | 60.16±0.32 | 53.2±6.1 | 92.68±2.86 | 29.56±5.07 | 82.76±7.34 | 63.94 | $2(M/K)DL$ |
| ICAE | 91.64±1.69 | 38.8±1.56 | 50.92±8.38 | 80.48±2.35 | 50.52±9.17 | 65.48±7.18 | 62.08±1.86 | 54.04±4.69 | 89.48±1.45 | 64.83 | $2(M/K)DL$ |
| CEPE | 74.28±3.9 | 36.2±0.56 | 55.48±3.42 | 78.00±3.49 | 59.12±1.6 | 61.72±5.26 | 87.24±1.2 | 42.28±3.31 | 82.36±1.61 | 64.08 | $2(M/K)DL$ |

**Experimental Setup** The following configuration is applied to all experiments unless otherwise specified. For each task, we randomly sample five demonstration examples per class[4] following the practice described in Wang et al. (2023) to avoid majority label bias (Zhao et al., 2021), and provide a fairly strong few-shot performance. No instruction is further applied to describe the task. Input sequences are formed using simple manually designed templates (included in Appendix A). For evaluation, we report the macro-average accuracy across nine tasks, computed under five random seeds. For the calibration process, we optimize linear coefficients for 100 epochs on the same demonstration set using the AdamW (Loshchilov & Hutter, 2019) optimizer. The learning rate starts at $1 \times 10^{-2}$ and anneals to $1 \times 10^{-5}$ according to a cosine scheduler. This calibration profile is applied uniformly across all architectures and tasks without tailoring.

## 3.1 BENCHMARKING I2CL

**I2CL Achieves Few-shot Performance at Zero-shot Inference Cost** I2CL is designed to enhance zero-shot performance, providing an alternative for ICL under constrained scenarios. As shown in Table 1, I2CL significantly outperforms the zero-shot counterpart by 17% in absolute accuracy and is only marginally behind (by around 1% on average) the few-shot learning. Note that I2CL consumes only zero-shot inference cost in terms of both memory usage and inference speed. An interesting observation arises from the Subj task, where the instructions (*i.e.*, demonstration examples) do not function effectively with ICL, yet are well adhered to under I2CL. We hypothesize that this phenomenon is due to the inherent properties of the pre-trained LLM, causing it to excel in certain tasks while lagging in others[5], and our proposed noisy self-calibration strategy can effectively rectify this deficiency.

**Comparison with Inference-time Methods w/o Additional Data** To further validate the efficacy of I2CL, we compare it with several comparable methods: (1) **Noise vector**: replacing the context vector with random noise to assess its necessity; (2) **Label-anchor**: a token reduction method from Wang et al. (2023), using formatting and label tokens as anchors; (3) **Task-vector**: task-vector (Hendel et al., 2023) also improves zero-shot performance without introducing additional computational and memory overhead at inference. It requires a hold-out validation set to identify the optimal replacement layer for each task at test-time; (4) **ICV**: we adapt In-context Vector Liu et al. (2024) method for our scenarios treating query tokens as negative examples and answer tokens as positive examples. We manually optimize ICV's strength parameter for each task and report the best performance. Please refer to Appendix B for implementation details.

As demonstrated in Table 1, replacing the context vector with random noise significantly degrades performance, highlighting the critical role of context vector. Label-anchor method exhibits a decent upgrade over the zero-shot baseline, achieving results most comparable to I2CL. Nevertheless, its inference cost remains dependent on the length of demonstration tokens, with reductions proportional to their compression rate. Like I2CL, the task-vector method also enjoys zero-shot inference expenses; yet its performance is sensitive to the downstream task and is, on average, slightly inferior to the

---

[4]One exception is DBPedia, where we use only one example per class due to the limitation of GPU memory.

[5]As demonstrated in Table 11, zero-shot performance of Subj is much higher under GPT-J than Llama2-7b though the later one is generally considered more powerful.

Table 2: Comparison between different PEFT-based few-shot fine-tuning strategies.

| Method | # trainable params. (K) ↓ | SST-2 (%) ↑ | SST-5 (%) ↑ | TREC (%) ↑ | AGNews (%) ↑ | Subj (%) ↑ | HateSpeech18 (%) ↑ | DBPedia (%) ↑ | EmoC (%) ↑ | MR (%) ↑ | Avg. acc. (%) ↑ |
|---|---|---|---|---|---|---|---|---|---|---|---|
| Prompt-tuning | 4.10 | 56.24±6.99 | 24.24±2.96 | 55.20±4.14 | 78.00±7.60 | 57.40±4.93 | 49.56±6.96 | 74.40±6.43 | 35.08±5.29 | 54.32±1.76 | 54.94 |
| LoRA | 4194.30 | 84.80±6.59 | 39.87±4.33 | 75.97±10.77 | 83.80±2.32 | 70.47±10.68 | 75.32±2.88 | 91.40±3.54 | 53.67±16.27 | 83.07±0.25 | 73.15 |
| IA3 | 262.14 | **89.40**±2.08 | **46.93**±0.81 | 75.41±4.94 | 84.43±1.45 | 56.67±3.07 | 62.54±5.58 | **93.91**±0.49 | 59.75±3.67 | **88.00**±1.88 | 73.00 |
| **I2CL (ours)** | **0.13** | 87.68±2.47 | 39.12±2.69 | **78.56**±5.32 | **85.48**±1.16 | **73.84**±3.84 | 69.88±5.67 | 90.16±1.86 | **63.72**±1.37 | 87.68±2.26 | **75.12** |

label-anchor method. Finally, a clear improvement over zero-shot baseline can be seen by adapting ICV method for few-shot classification tasks. Notwithstanding the boost, it necessitates manual effort for hyperparameter selection, which limits its applicability for scenarios that favor automated pipelines. Compared to the above methods, I2CL achieves the best performance on all tasks with neither manual intervention nor task-specific hyperparameter selection.

**Comparison with Inference-time Prompt Compression Methods** Another line of work that leverages external large-scale datasets to learn an amortized context compressor can also be applied to ICL scenarios. To this end, we compare I2CL with three SoTA prompt compression methods: **AutoCompressors** Chevalier et al. (2023), **ICAE** Ge et al. (2023), and **CEPE** Yen et al. (2024). Here, we directly apply their released pre-trained models on our tasks to avoid implementation bias. As shown in the bottom section of Table 1, all three methods yield a decent im-

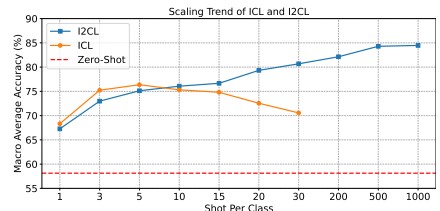

Figure 3: Scaling trend of I2CL.

provements over zero-shot baseline, demonstrating their effectiveness in compressing context tokens. However, these methods produce mixed and fluctuating results across different tasks, likely due to their different training data and strategies. Instead of learning a universal compressor, I2CL opts for customizing a task-specific compression strategy at test time, rendering it additional adaptability for few-shot learning tasks. Nevertheless, we do not claim overall superiority of I2CL over prompt-compression methods as they are deliberately tailored for excessively long context compression which is orthogonal to the primary application of I2CL.

**Comparison with Test-time PEFT Methods.** The design of I2CL involves a lightweight training process (*i.e.*, self-calibration). As such, we further establish a comparison between I2CL and three representative PEFT methods: prompt-tuning (Lester et al., 2021), LoRA (Hu et al., 2021), and IA3 (Liu et al., 2022) to underscore the effectiveness and efficiency of I2CL. As shown in Table 2, prompt-tuning fails under such data scarcity, while LoRA and IA3 perform on par with I2CL. Importantly, I2CL achieves the best overall performance with approximately 100x fewer learnable parameters than prompt-tuning, 1,000x fewer than IA3, and 10,000x fewer than LoRA. We refer readers to Appendix B for implementation details.

**Scaling Property of I2CL** Although I2CL is primarily designed for few-shot scenarios, it exhibits strong scaling properties. As shown in Fig. 3, ICL peaks at around 5-shots and additional demonstration examples do not necessarily bring further benefits. In contrast, I2CL can easily scale up to hundreds or even thousands of demonstration examples without performance degradation, and it readily surpasses few-shot performance under a modest number of demonstration examples. We note that using more demonstration examples increases the calibration cost.

## 3.2 ON THE FORMATION OF THE CONTEXT VECTOR

Context vector plays a fundamental role in the design of I2CL. Here, we conduct an in-depth analysis to investigate its properties and the factors affecting its functionality.

**Deficient ICL ≠ Deficient I2CL** It is well-known that ICL is sensitive to the choice of demonstration examples (Dong et al., 2023), even their order (Zhao et al., 2021; Lu et al., 2022). In this context, we investigate whether poorly performing demonstration examples in ICL will similarly affect I2CL. We sample 20 groups of demonstrations with random instances and orders, and identify the group with the poorest ICL performance on a hold-out dataset that is non-overlapping with both train and evaluation sets. Using these same demonstration examples, we perform both ICL and I2CL on the evaluation set. Referencing Figure 4 (left), the poorly performing demonstrations lead to a severely degraded ICL performance (−7%), while I2CL performance remains largely unaffected (−0.5%). This result suggests that I2CL requires less careful curation of demonstration examples than ICL. We hypothesize that this robustness stems from I2CL's ability to extract task-relevant features that

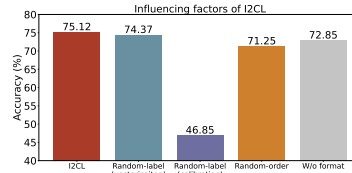 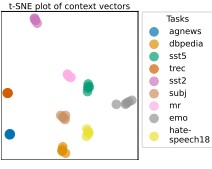

Figure 4: **Left**: Evaluation of I2CL and few-shot learning under deficient demonstrations. The symbol ∗ denotes the results under deficient demonstration examples. "Unseen demo" refers to the evaluation of calibrated coefficients on unseen demonstrations. **Middle**: Analysis of the influencing factors of context vectors. "Random-label" indicates random input-label mappings. "Random-order" refers to the random permutation of words. "W/o format" signifies excluding the template tokens during the creation of context vectors. **Right**: t-SNE plot of context vectors. Each circle denotes a context vector generated using a group of randomly sampled demonstration examples.

are invariant to surface-level variations in the demonstrations. To further corroborate this hypothesis, we visualize the context vectors using t-SNE (van der Maaten & Hinton, 2008) in Figure 4 (right). The visualization reveals that context vectors from different demonstration groups cluster tightly by task, indicating that I2CL captures consistent task-specific representations despite variations in the demonstrations.

**Influencing Factors of the Context Vector Generation**    We have shown above that context vectors are robust against the variations in demonstration examples. Here, we explore various factors that may impact the context vector's functionality. Inspired by counter-intuitive findings reported in (Zhang et al., 2022; Min et al., 2022b), we experiment with two variations of demonstration examples. **Random-label**: pairing each input sentence with a random label from the task, rather than the ground-truth label. **Random-token**: randomly permuting all words within a demonstration example. Another ineligible factor involves the inclusion of formatting tokens. To this end, we also evaluate on **W/o-format**: removing formatting words from a demonstration example, retaining only the input sentence and its corresponding label, *e.g.*, "~~Review:~~ This is a great movie. ~~Label:~~ positive." As revealed in Figure 4 (middle), input-label mapping relations have minimal impact on context vector formation, mirroring the observations made in ICL. However, these mapping relations are crucial for calibration purposes; using random input-label mapping during calibration undermines the functionality of I2CL. Unlike the phenomenon observed in (Zhang et al., 2022), word sequence holds essential statistics under a causal architecture, and randomly permuting input words yields degenerated context vectors, leading to a clear performance drop. Lastly, formatting tokens contribute substantially—removing them results in a noticeable performance degradation.

## 3.3    ANALYSIS OF CALIBRATED LINEAR COEFFICIENTS

With a grasp on the formation of context vectors, we delve into the properties of calibrated linear coefficients to enhance our understanding of the internal mechanisms of I2CL.

**Linear Coefficients Are Generalizable**    Given the demonstration-dependent context vectorization and calibration process, it is natural to consider I2CL as an inference-time optimization framework. However, we demonstrate that the coefficients are generalizable and require calibration only once per task. Concretely, we evaluate a set of calibrated coefficients on five new context vectors generated using five unseen groups of demonstrations. As exhibited in Figure 4 (left), the calibrated linear coefficients generalize well to unseen demonstrations. We attribute this generalization capability to two factors: (1) the inherent robustness of the context vector against token space variations, as validated in Section 3.2, and (2) the sufficiency of calibrated linear coefficients, independent of context vectors and query activation, to uniquely represent a task. We substantiate the second premise in the following subsection.

**Calibrated Coefficients Embed Task Semantics**    Here, we investigate whether calibrated coefficients alone are adequate to serve as task-ids. In pursuit of this goal, we concatenate calibrated coefficients across layers to form a one-dimensional vector, which we then visualize using t-SNE. As illustrated in Figure 5 (left), the calibrated linear coefficients are tightly clustered for instances associated with the same task and fall apart otherwise, providing complementary evidence for their good generalization capability. One exception comes from the proximity between MR and SST-2. This phenomenon is not unexpected, as both tasks originate from the same underlying distribution (*i.e.*, rotten tomatoes movie reviews), suggesting that similarities among calibrated coefficients may

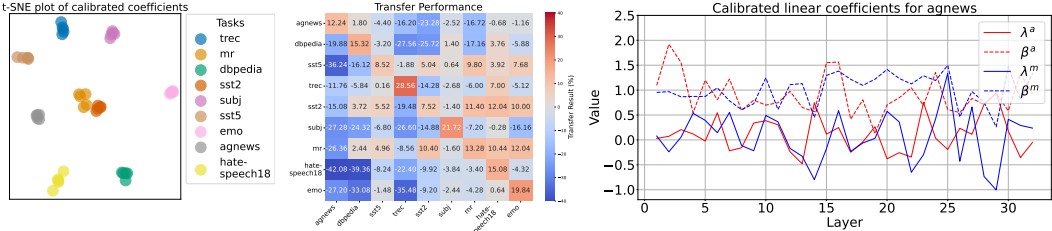

Figure 5: **Left**: t-SNE visualization of calibrated linear coefficients. Each circle denotes a runtime with a random seed. **Middle**: This image displays the transfer results among various tasks. Each row represents a source task and each column denotes a target task. Red and blue colors signify positive and negative transfer outcomes, respectively. **Right**: This plot shows the calibrated linear coefficients for SST-2. $\lambda^a, \beta^a, \lambda^m, \beta^m$ are the layer-wise coefficients described in Equation 4.

indicate potential transferability among tasks. To explore this further, we transfer the context vectors and calibrated linear coefficients from a source task to various target tasks. We then measure the differences between the transferred results and their respective zero-shot outcomes to identify both positive and negative transfers. Figure 5 (middle) shows that performance on MR can be effectively enhanced by transferring context vectors and calibrated coefficients from SST-2, and vice versa. Similarly, both SST-2 and MR benefit from transfers from SST-5, likely due to their shared focus on sentiment analysis. An intriguing aspect of our findings is the asymmetry of transferability among different tasks; a successful transfer in one direction does not necessarily forecast the opposite (e.g., transfer between HateSpeech18 and SST-2).

**Enhance New Task with Existing Anchors** I2CL posits two pivotal features distinct from typical ICL: (1) An interface that facilitates vector arithmetic; (2) A set of linear coefficients that act as task-ids, laying a foundation for transfer learning (Zhuang et al., 2021). Here, we present an transfer learning algorithm based on these features. Let $c_1, \ldots, c_N$ represent calibrated coefficients for existing tasks, and $c_{\text{new}}$ for the new task. Associated with these coefficients, respectively, are context vectors $v_1, \ldots, v_N$, and $v_{\text{new}}$. We compute the cosine similarity between $c_{\text{new}}$ and each $c_i$ for $i = 1, \ldots, N$ and retain indices $I = \{i \mid \cos(c_{\text{new}}, c_i) > h\}$ according to a pre-defined threshold $h$. The cosine similarities for the indices in $I$ are then converted into a probability distribution using the softmax function:

Table 3: Transfer learning result. Only tasks having more than one similar task (according to $h$) in our task curation are exhibited.

|  | I2CL | |
|---|---|---|
| Task | W/o transfer (%) | W/ transfer (%) |
| SST-5 | $39.12_{\pm 2.69}$ | $\mathbf{43.24}_{\pm 3.70}$ |
| MR | $87.68_{\pm 2.47}$ | $\mathbf{89.99}_{\pm 2.83}$ |

$$P(i) = \frac{\exp(\cos(c_{\text{new}}, c_i)/\tau)}{\sum_{j \in I} \exp(\cos(c_{\text{new}}, c_j)/\tau)}, \quad \forall i \in I, \qquad (7)$$

where $\tau$ is the temperature. Finally, we reinitialize $v_{\text{new}}$ and $c_{\text{new}}$ using the weighted average of retained context vectors and coefficients: $v_{\text{avg}} = \sum_{i \in I} P(i)v_i$, $c_{\text{avg}} = \sum_{i \in I} P(i)c_i$, and perform another round of calibration. We refer to Appendix B for detailed algorithms and implementation. Empirical results in Table 3 demonstrate a clear benefit of the proposed transfer learning method.

**Injection Dynamics** Thus far, we have demonstrated that a few static scalars, conditioned on appropriate context vectors, can successfully execute a wide range of tasks. Herein, we delve into how context vectors are injected. Using the AGNews as an example, we examine the coefficient values across different layers (additional plots in Appendix C.4). One reasonable speculation postulates a gradual addition of context vectors to the residual streams. Nevertheless, observations from Figure 5 (right) reveal a more nuanced control over the information injection process. Instead of monotonically adding (+) more information to the residual streams, I2CL also allows the deletion (−) of information at certain layers, with coefficient values fluctuating across different layers.

### 3.4 ABLATION STUDY

In this section, we conduct a comprehensive ablation study on the module, layer, and token position of injection, as well as the noise scale. We also explore various vectorization and injection formulas to highlight the rationale of our designs.

**Target Module** I2CL extracts and injects activations at both **MHA** and **MLP** modules. To identify the contribution of each module, we test using either MHA or MLP independently, and we also

Table 4: Target modules.

| Name | Accuracy (%) |
|---|---|
| Zero-shot | 58.13 |
| MHA | 66.97 |
| MLP | 70.27 |
| Hidden state | 56.80 |
| MHA+MLP (ours) | 75.12 |

Table 5: Target layers.

| Name | Accuracy (%) |
|---|---|
| Zero-shot | 58.13 |
| Early | 58.18 |
| Middle | 64.07 |
| Late | 64.03 |
| All (ours) | 75.12 |

Table 6: Injection position.

| Name | Accuracy (%) |
|---|---|
| Zero-shot | 58.13 |
| Random | 59.86 |
| First | 62.14 |
| Last | 66.75 |
| All (ours) | 75.12 |

Table 7: Noise scale.

| Name | Accuracy (%) |
|---|---|
| Zero-shot | 58.13 |
| $\gamma = 0.0$ | 72.23 |
| $\gamma = 0.01$ | 40.53 |
| $\gamma = 0.001$ (ours) | 75.12 |

Table 8: Injection formula.

| Name | Accuracy (%) |
|---|---|
| Zero-shot | 58.13 |
| $\lambda \boldsymbol{v} + \boldsymbol{a}, \lambda > 0$ | 63.63 |
| $(\lambda \boldsymbol{v} + (1 - \lambda)\boldsymbol{a}) \times \beta, \beta > 0$ | 71.39 |
| $\lambda \boldsymbol{v} + \beta \boldsymbol{a}$ (ours) | 75.12 |

consider leveraging the **hidden state** at each layer. As shown in Table 4, extracting and injecting context vectors at either MHA or MLP proves beneficial, with MLP showing a clear advantage. We hypothesize this is due to the additional engagement of information stored in the MLP weights. However, targeting the hidden state does not lead to any improvement, likely due to the accumulation effect which complicates the optimization process within self-calibration stage.

**Target Layer**    I2CL encompasses all layers in a large language model (LLM), eliminating the need for specifying layer indexes. This model-agnostic approach not only simplifies the setup but also enhances performance, as evidenced in Table 4. We divide the model into three sub-parts—**early**, **middle**, and **late**—each containing one-third of the total layers. We then apply I2CL only within the target layer range. The middle and late layers prove more effective than the early layers, and injecting across all layers provides a clear performance boost, highlighting the importance of fusing information at all levels of abstraction.

**Injection Position**    By default, I2CL injects context vectors into all residual streams during inference. To justify this choice, we test injections only at **random**, **first**, and **last** residual streams. As shown in Table 6, and aligning with common intuition, injecting at the end residual stream yields the largest improvement compared to other positions, although it still shows a significant gap compared to injecting at all residual streams.

**Noise Scale**    We evaluate the impact of noise strength during the calibration phase. As demonstrated in Table 7, an appropriate noise scale leads to a clear performance gain. Conversely, an improper strength can disrupt the propagation of information at inference, resulting in deteriorated outcomes. We empirically identify $\tau = 0.001$ as a suitable scale and have applied it across all tasks and architectures.

**Injection Formula**    I2CL utilizes a linear combination to blend context vectors with query activations. Let $\boldsymbol{v}$ denote the context vector and $\boldsymbol{a}$ indicates activation; we use $\lambda \boldsymbol{c} + \beta \boldsymbol{a}$. One simplification is to view the injection process as solely adding the context vector to the activation: $\lambda \boldsymbol{v} + \boldsymbol{a}$ with $\lambda > 0$ as done in Liu et al. (2024). Another common formula involves constraining the sum of linear coefficients to one and allowing a separate scale factor: $(\lambda \boldsymbol{v} + (1 - \lambda)\boldsymbol{a}) \times \beta, \beta > 0$. According to Table 8, a linear combination with no constraints achieves the best overall performance. Therefore, we conjecture that it is critical to allow not only information addition but also deletion, *i.e.*, permitting a negative sign for the linear coefficient, and to scale each vector independently. Observations in Figure 5 (right) corroborate this conjecture.

# 4    RELATED WORK

**Understanding In-context Learning**    Besides enhancing ICL, the exploration of ICL's internal mechanisms has attracted significant research attention. Akyürek et al. (2023) draw parallels between ICL and gradient descent in linear regression tasks, suggesting a fundamental alignment with classical optimization methods. Complementary perspectives from Von Oswald et al. (2023) and Dai et al. (2023) conceptualize ICL as a form of meta-optimization, further enriching our understanding of its operational basis. Concurrently, Xie et al. (2022) interpret ICL through the lens of implicit Bayesian inference, proposing a probabilistic foundation for the learning process. Wei et al. (2023)

and Olsson et al. (2022) respectively attribute ICL's capabilities to the establishment of input-label correspondences and the identification of so-called induction heads, highlighting the intricate interplay between data representation and model interpretability. Most recently, Label-as-Anchors (Wang et al., 2023) inspects ICL from an information flow perspective and leverages anchor tokens to perform token reduction. Similarly, I2CL also enhances the efficiency of ICL, but distinguishes itself by circumventing the need for caching latents of anchor tokens or employing multi-head attention, thereby reducing the inference cost to that of the zero-shot.

**Activation/representation Engineering** An emerging research field, termed activation/representation engineering, closely relates to our study. Recent endeavors by Merullo et al. (2023); Turner et al. (2023); Zou et al. (2023) have unveiled the phenomenon of steering vectors, which can be derived from a positive and negative text pair and used to steer the content generation of LLMs. Steering vectors can also be learned via gradient descent (Subramani et al., 2022). Liu et al. (2024) applies these insights to bolster LLM safety while Li et al. (2023) explores their utility in eliciting more truthful responses from LLMs. Steering vector has also been adapted to multi-modal domain to reduce hallucination (Li et al., 2025). To better understand the inner mechanism of activation engineering, the linear representation hypothesis has been studied and discussed in Li et al. (2021); Hernandez et al. (2024); Park et al. (2023). Central to this discourse, the idea of task/function vector (Hendel et al., 2023; Todd et al., 2024) resonates with the core premise of I2CL. Both methodologies extract task/function vectors from the demonstration examples and use them to improve zero-shot performance. I2CL stands out not only due to its superior performance, but also through its simplicity, namely, its avoidance of paired demonstrations, and the need for task- or architecture-specific hyperparameter tuning, such as selecting attention heads through causal mediation or determining target layers via extra validation sets.

**Prompt Compression for LLMs** Prompt compression (Chang et al., 2024) is also related to our work, as it shares a similar goal of improving the inference efficiency of LLMs. However, prompt compression methods emphasize on reducing the length of excessive contexts by learning an additional amortized compressor (Ge et al., 2023; Chevalier et al., 2023; Yen et al., 2024), while I2CL transforms the standard few-shot learning approach into a zero-shot manner at test time, treating the LLM itself as a context vector generator. Most prompt compression methods operate by compressing a long prompt sequence into a set of compact soft prompts, leveraging the attention mechanism to further propagate context information. In contrast, I2CL generates a context vector from the activation space and intervenes directly in the residual streams.

## 5 LIMITATIONS

I2CL is subject to several limitations. First, we confine the scope of this initial exploration to standard text classification tasks, leaving more sophisticated tasks for future research. It is non-trivial to further extend I2CL to the realm of open-ended generation tasks and those involving multi-hop reasoning processes. Second, I2CL necessitates access to and caching of intermediate activations from language models, which may not be feasible with state-of-the-art commercial models (e.g., GPT-4, Gemini, Claude3). Thirdly, limited by the computational resources, we evaluated I2CL on modest-sized models, and further scaling the evaluation to commercial size models could yield additional insights.

## 6 CONCLUSION

In this study, we introduce Implicit In-context Learning (I2CL), a simple and novel framework that integrates a minimal set of demonstration examples within the activation space of LLMs. Diverging from ICL, I2CL eliminates the need for caching the latents of demonstration tokens and replaces the non-linear information fusion process (*i.e.*, attention head) with linear operations. Therefore, I2CL reduces both computational and memory expenses during inference to that of zero-shot level. Moreover, I2CL is validated to be robust against token space variations, and it facilitates a novel representation of task-ids which enhances task similarity detection and fosters transfer learning. Empirical evidence on nine real-world tasks across three different models suggests the potential of I2CL as a more efficient and robust alternative to ICL for constrained scenarios. Through a set of in-depth analyses and ablations, we also shed light on the internal working mechanics of I2CL.

ACKNOWLEDGMENTS

We extend our sincere gratitude to Prof. Diyi Yang for her constructive suggestions on the motivation and empirical design of this study. Her expertise and insights were invaluable to this research.

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

## APPENDIX CONTENTS

## A  PROMPTING TEMPLATES

Table 9: Prompting templates and label spaces used in our experiments. {Sentence} and {Label} are placeholders for the input sentence and its corresponding label. We exhibit only the template of a single example for the illustration purpose, and multiple demonstration examples are connected by a newline character: '\n'.

| Dataset | Template | Label Space |
|---|---|---|
| SST-2 | Review: {Sentence}
Sentiment: {Label} | negative / positive |
| SST-5 | Sentence: {Sentence}
Sentiment: {Label} | terrible / negative / neutral / positive / great |
| MR | Review: {Sentence}
Sentiment: {Label} | negative / positive |
| Subj | Sentence: {Sentence}
Label: {Label} | objective / subjective |
| DBPedia | Input: {Sentence}
Label: {Label} | company / school / artist / athlete / politics / transportation / building / nature / village / animal / plant / album / film / book |
| AGNews | News: {Sentence}
Type: {Label} | World / Sports / Business / Technology |
| TREC | Question: {Sentence}
Answer Type: {Label} | Abbreviation / Entity / Person / Location / Number |
| HateSpeech18 | Text: {Sentence}
Label: {Label} | neutral / hate |
| EmoC | Dialogue: {Sentence}
Emotion: {Label} | others / happy / sad / angry |

**Extra Details**    For the task HateSpeech18, we preprocess the data to retain only the first two classes—{0: neutral} and {1: hate}. We exclude the other two classes due to their extremely limited number of samples.

# B    REPRODUCTION DETAILS

## B.1    IMPLEMENTATION OF BASELINE METHODS.

**Noise Vector**. In this baseline method, We simply replace context vectors with random noises while keeping all other settings identical to I2CL.

**Label Anchor** (Wang et al., 2023). We take the officially released code, aligning the architecture, datasets, and template setups for a fair comparison. Following their established practice, template tokens and the newline separator '\n' are used as anchors. Detailed information can be found at `https://github.com/lancopku/label-words-are-anchors`.

**Task Vector** (Hendel et al., 2023). We replicate the task vector method which was initially evaluated on a set of toy datasets. To generate the task vector, we append a random extra query after the concatenated demonstration examples (five per class) to simulate the dummy query they utilized. We then extract the hidden state from the token position responsible for the next token prediction to serve as the task vector. A hold-out dataset with 32 extra examples is used to select the best layer for extraction and replacement for each task, following the original method.

**In-contex Vector** (Liu et al., 2024). ICV method is primarily crafted for open-end generation tasks, and it leverages positive and negative demonstration pairs to generate the steering vector. In our scenario, we set sentence (or question) in a demonstration example as negative and its corresponding answer as positive counterpart to extract in-context vector. We then manually search the strength scalar of injection for each task. Specifically, we first search in a log scale covering $\lambda \in [0.0001, 0.001, 0.01, 0.1, 1.0]$, followed by a more fine-grained search between $0.01$ and $0.1$. The best strength scalars we used are reported in Table 10. We refer readers to `https://github.com/shengliu66/ICV` for more technique details.

Table 10: Strength scalar $\lambda$ for ICV.

| Task | $\lambda$ |
|---|---|
| SST-2 | 0.01 |
| SST-5 | 0.02 |
| TREC | 0.02 |
| AGNews | 0.0001 |
| Subj | 0.02 |
| HateSpeech18 | 0.02 |
| DBPedia | 0.001 |
| EmoC | 0.02 |
| MR | 0.02 |

**AutoCompressors** (Chevalier et al., 2023). We directly test the officially released model from here: `https://github.com/princeton-nlp/AutoCompressors` on our tasks. All hyper-parameters are set by default values. Evaluation protocols remain the same as ours.

**ICAE** (Ge et al., 2023). We directly test the officially released model from here: `https://github.com/getao/icae` on our tasks. All hyper-parameters are set by default values. Evaluation protocols remain the same as ours.

**CEPE** (Yen et al., 2024). We directly test the officially released model from here: `https://github.com/princeton-nlp/CEPE` on our tasks. All hyper-parameters are set by default values. Evaluation protocols remain the same as ours.

**Prompt-tuning** (Lester et al., 2021). For the implementation of prompt-tuning method, we allow one extra learnable token ($P = 1$) per layer, and apply learnable prompts across all layers. We also attempt to use more learnable tokens, resulting in poorer performance due to overfitting. For optimization, we conduct a simple grid search on SST-2 to determine an optimal learning rate of $0.1$, which we then apply across all tasks. All other configurations remain as specified in the experimental section.

**LoRA** (Hu et al., 2021) and **IA3** (Liu et al., 2022). We use implementations from HuggingFace PEFT library for both PEFT methods. Learning rate is set to $0.001$ for both methods, and all other optimization protocols are kept the same as in experimental section. The LoRA configuration uses a rank of $8$ with a scaling factor ($\alpha$) of $32$, applies dropout at a rate of $0.05$, and targets both the query and value projection modules. As for IA3 method, adaptation is applied not only on query and value projection modules, but also for feed-forward layers.

Table 11: Comparison between I2CL and baseline methods on GPT-J-6B.

| Task | Zero-shot | Few-shot (ICL) | Soft-prompt | Label-anchor | Task-vector | **I2CL (ours)** |
|------|-----------|----------------|-------------|--------------|-------------|-----------------|
| SST-2 (%) ↑ | 77.76 | 89.44±2.60 | 69.04±11.61 | 87.12±4.57 | 59.84±4.47 | 85.48±1.18 |
| SST-5 (%) ↑ | 25.60 | 39.65±4.57 | 37.88±2.99 | 37.24±3.53 | 31.20±2.82 | 37.32±3.11 |
| TREC (%) ↑ | 68.20 | 67.76±2.11 | 67.00±12.04 | 58.52±2.44 | 67.32±0.32 | 63.84±7.58 |
| AGNews (%) ↑ | 71.60 | 83.18±2.03 | 83.16±4.86 | 80.84±0.88 | 80.12±2.23 | 81.56±3.13 |
| Subj (%) ↑ | 62.40 | 50.20±0.22 | 63.64±6.52 | 51.16±1.71 | 66.32±2.31 | 65.56±8.33 |
| HateSpeech18 (%) ↑ | 59.92 | 53.44±6.84 | 67.76±5.04 | 55.20±8.19 | 70.12±5.04 | 62.32±5.76 |
| DBPedia (%) ↑ | 65.56 | 93.30±1.19 | 85.04±1.02 | 90.84±1.79 | 77.64±4.63 | 81.84±4.50 |
| EmoC (%) ↑ | 44.68 | 47.62±8.62 | 47.48±10.87 | 44.00±8.25 | 45.00±4.20 | 50.32±4.68 |
| MR (%) ↑ | 76.88 | 88.66±1.23 | 73.76±9.40 | 87.92±3.82 | 81.36±3.58 | 84.40±2.45 |
| Macro avg. acc. (%) ↑ | 61.40 | 68.14 | 66.08 | 65.87 | 64.32 | 68.07 |

## B.2 Implementation of Transfer Learning Method

Algorithm 1 details the transfer learning method proposed for I2CL. In implementation, we empirically set $h = 0.8$ and $\tau = 0.5$.

---

**Algorithm 1** Transfer Learning of I2CL

---

1: **Input:** Coefficients $c_1, c_2, \ldots, c_N$, Context vectors $v_1, v_2, \ldots, v_N$, Demonstrations of new task $d_{new}$, Threshold $h$, Temperature $\tau$, Default coefficient initialization $c_{init}$.
2: **Output:** $c_{new}, v_{new}$
3: Initialize $I \leftarrow \emptyset$
4: $v_{new} \leftarrow \text{Context\_vectorization}(d_{new})$
5: $c_{new} \leftarrow \text{Noisy\_self\_calibration}(d_{new}, v_{new}, c_{init})$
6: **for** $i = 1$ to $n$ **do**
7:   Compute $s_i \leftarrow \text{cosine}(c_{new}, c_i)$
8:   **if** $s_i > h$ **then**
9:     Add $i$ to $I$
10:   **end if**
11: **end for**
12: Compute probabilities $P(i) \leftarrow \frac{\exp(s_i)/\tau}{\sum_{j \in I} \exp(s_j/\tau)}$ for each $i \in I$
13: Compute $v_{avg} \leftarrow \sum_{i \in I} P(i) v_i$
14: Compute $c_{avg} \leftarrow \sum_{i \in I} P(i) c_i$
15: $c_{new} \leftarrow \text{Noisy\_self\_calibration}(d_{new}, v_{avg}, c_{avg})$
16: $v_{new} \leftarrow v_{avg}$
17: **return** $c_{new}, v_{new}$

---

## C Additional Experiments and Analysis

### C.1 Results under GPT2-XL, GPT-J, and Llama3-8B

Here, we further compare I2CL with baseline methods that do not need manual intervention or additional datasets on other two popular architectures. The results under architecture GPT-J and GPT2-XL are shown in Table 11 and Table 12, respectively. To highlight the generalization ability of I2CL, we further extend I2CL to latest Llama3-8B model and establish a comparison between zero-shot and few-shot learning in Table 13.

### C.2 Analysis on Synthetic Dataset

To underscore the generality of the proposed I2CL, we crafted a synthetic dataset that containing no semantic and formatting priors. Concretely, we generated three types of data containing "**random strings**", "**random special characters**", and "**random numerical values**", labeled as "**A**" "**B**" and "**C**", respectively. We then evaluate the performance of zero-shot, few-shot and I2CL on this synthetic dataset following the same setting as for main Table 1. As shown in Table 14, I2CL outperforms

Table 12: Comparison between I2CL and baseline methods on GPT2-XL. AGnews and DBPedia are not evaluated due to the limitation of GPT2-XL's context window size.

| Task | Zero-shot | Few-shot (ICL) | Soft-prompt | Label-anchor | Task-vector | **I2CL (ours)** |
|---|---|---|---|---|---|---|
| SST-2 (%) ↑ | 74.76 | 73.65$_{\pm 8.89}$ | 61.04$_{\pm 3.45}$ | 63.40$_{\pm 8.82}$ | 81.08$_{\pm 4.87}$ | 80.16$_{\pm 3.98}$ |
| SST-5 (%) ↑ | 30.44 | 35.95$_{\pm 2.39}$ | 23.96$_{\pm 2.09}$ | 22.36$_{\pm 3.37}$ | 28.52$_{\pm 1.37}$ | 33.84$_{\pm 2.60}$ |
| TREC (%) ↑ | 35.40 | 60.64$_{\pm 5.00}$ | 40.60$_{\pm 10.15}$ | 66.36$_{\pm 10.69}$ | 41.40$_{\pm 5.35}$ | 51.48$_{\pm 5.26}$ |
| Subj (%) ↑ | 64.88 | 63.82$_{\pm 10.55}$ | 55.44$_{\pm 4.12}$ | 55.56$_{\pm 4.26}$ | 71.80$_{\pm 1.86}$ | 65.96$_{\pm 4.83}$ |
| HateSpeech18 (%) ↑ | 70.84 | 51.86$_{\pm 3.22}$ | 63.92$_{\pm 7.06}$ | 54.88$_{\pm 4.53}$ | 62.48$_{\pm 2.83}$ | 68.32$_{\pm 4.76}$ |
| EmoC (%) ↑ | 37.88 | 38.62$_{\pm 7.68}$ | 33.60$_{\pm 4.04}$ | 36.68$_{\pm 2.70}$ | 37.60$_{\pm 2.48}$ | 47.92$_{\pm 1.84}$ |
| MR (%) ↑ | 71.36 | 75.79$_{\pm 9.25}$ | 57.60$_{\pm 3.53}$ | 60.20$_{\pm 3.32}$ | 78.40$_{\pm 2.36}$ | 83.20$_{\pm 3.29}$ |
| Macro avg. acc. (%) ↑ | 55.08 | 57.19 | 48.02 | 51.35 | 57.33 | 61.55 |

Table 13: Performance comparison between Zero-shot, ICL, and I2CL on Llama3-8B.

| Name | SST-2 | SST-5 | TREC | AGNews | Subj | HateSpeech18 | DBPedia | EmoC | MR | Avg. |
|---|---|---|---|---|---|---|---|---|---|---|
| Zero-shot | 55.60 | 31.40 | 66.40 | 73.60 | 51.00 | 50.40 | 56.20 | 41.00 | 55.60 | 53.47 |
| ICL | 93.00 ± 0.62 | 39.4 ± 3.21 | 78.2 ± 5.94 | 84.8 ± 1.30 | 62.47 ± 12.44 | 64.93 ± 2.65 | 82.33 ± 3.64 | 52.20 ± 3.92 | 92.73 ± 0.50 | 81.26 |
| I2CL | 91.40 ± 2.26 | 32.60 ± 1.25 | 80.27 ± 2.58 | 82.56 ± 1.57 | 64.67 ± 3.52 | 75.80 ± 1.02 | 85.00 ± 0.33 | 52.92 ± 5.28 | 84.27 ± 2.07 | 81.18 |

few-shot learning (*i.e.*, ICL) by a large margin, highlighting the effectiveness and generality of I2CL, and we attribute this improvement to the proposed noisy self-calibration and vector injection methods which directly intervene at residual streams. Similar to the empirical observation in Sec. 3.3, the calibrated linear coefficients are also generalizable, and they can be directly used under unseen demonstration examples without additional calibration (see last column in Table 14).

## C.3 APPLY I2CL OVER ICL

As I2CL and ICL are not mutually exclusive, an interesting empirical exploration involves applying I2CL on top of ICL. Specifically, we retain the demonstration tokens in the context throughout the noisy self-calibration and inference stages. As shown in Table 15, the combination of I2CL with ICL significantly improves performance on certain tasks, surpassing ICL by a substantial margin. Moreover, for tasks where I2CL originally underperforms compared to ICL, the inclusion of demonstration tokens helps bridge the gap. These empirical observations highlight the versatility of I2CL. On one hand, I2CL can serve as a substitute for ICL to reduce inference costs; on the other hand, when computational and memory constraints are less critical, I2CL can be applied in conjunction with ICL to further enhance ICL's performance.

## C.4 ADDITIONAL VISUALIZATION

Here, we present in Fig. 6 the calibrated coefficients for all tasks we used, illustrating variations and patterns across different configurations.

## C.5 VISUALIZATION OF IN-TASK CONTEXT VECTORS

As demonstrated in Fig. 4 (right), context vectors embed task semantics and context vectors from different classes are well separated. However, it is unclear whether context vector can carry label information within a more nuanced in-task perspective. To this end, we extract class-wise context vectors within each task and visualize in-task relations among different context vectors in Fig. 7. As shown in the figures, context vectors from different classes are well-separated in most tasks, indicating that context vectors will also carry label information. A notable exception comes from EmoC where vectors appear more mixed. We conjecture that besides label information, a substantial inherent parametric knowledge has also been hubbed by the context vector.

Table 14: Evaluation of zero-shot, few-shot and I2CL on the synthetic dataset.

| Task | Zero-shot | Few-shot (ICL) | I2CL | I2CL (unseen demo.) |
|---|---|---|---|---|
| Synthetic data | 32.6 | 66.20$_{\pm 0.73}$ | **86.48**$_{\pm 4.51}$ | 86.36$_{\pm 5.40}$ |

Table 15: Results of applying I2CL on top of ICL. Task AGNews is not applicable under 5-shot due to the limitation of GPU memory.

| Name | SST-2 | SST-5 | TREC | Subj | HateSpeech18 | DBPedia | EmoC | MR | Average |
|---|---|---|---|---|---|---|---|---|---|
| ICL | **94.44**$_{\pm 1.44}$ | 41.72$_{\pm 3.68}$ | 77.32$_{\pm 4.41}$ | 52.56$_{\pm 3.09}$ | 70.24$_{\pm 5.80}$ | **96.64**$_{\pm 0.48}$ | **75.48**$_{\pm 1.63}$ | **93.24**$_{\pm 0.50}$ | 75.21 |
| I2CL | 87.68$_{\pm 2.47}$ | 39.12$_{\pm 2.69}$ | 78.56$_{\pm 5.32}$ | 73.84$_{\pm 3.84}$ | 69.88$_{\pm 5.67}$ | 90.16$_{\pm 1.86}$ | 63.72$_{\pm 1.37}$ | 87.68$_{\pm 2.26}$ | 73.83 |
| ICL+I2CL | 93.52$_{\pm 0.39}$ | **44.53**$_{\pm 1.61}$ | **83.56**$_{\pm 5.18}$ | **89.84**$_{\pm 4.09}$ | **81.36**$_{\pm 1.24}$ | 95.87$_{\pm 0.62}$ | 73.84$_{\pm 6.36}$ | 93.16$_{\pm 1.01}$ | **81.96** |

# D  ADDITIONAL DISCUSSION

## D.1  POTENTIAL DESIGN SPACE

Although the end residual stream contains a rich repository of information for the given token sequence, other token positions may also possess valuable attributes, as noted in recent studies (Hendel et al., 2023; Todd et al., 2024; Liu et al., 2024). Recent research also highlights the importance of formatting tokens during information propagation (Wang et al., 2023; Bai et al., 2024). We believe our approach could benefit from a more sophisticated design of demonstration vectorization. Additionally, while we currently use a single scalar to gauge the strength of aggregated multi-head attention, allowing finer granularity—such as separate strength scalars for different attention heads—might enhance our system's performance, albeit at the cost of increased parameters.

## D.2  BROADER IMPACTS

I2CL inherits the same risks as standard In-context Learning. While I2CL primarily serves to enhance efficiency, its ability to adapt quickly to different data could potentially be used for creating misinformation or other harmful content at scale. I2CL's reliance on existing data for demonstration examples could propagate existing biases if the data are not carefully curated. One additional risk comes from the generation of implicit vector representation for the demonstration examples, making the detection of the malicious contents even more challenging.

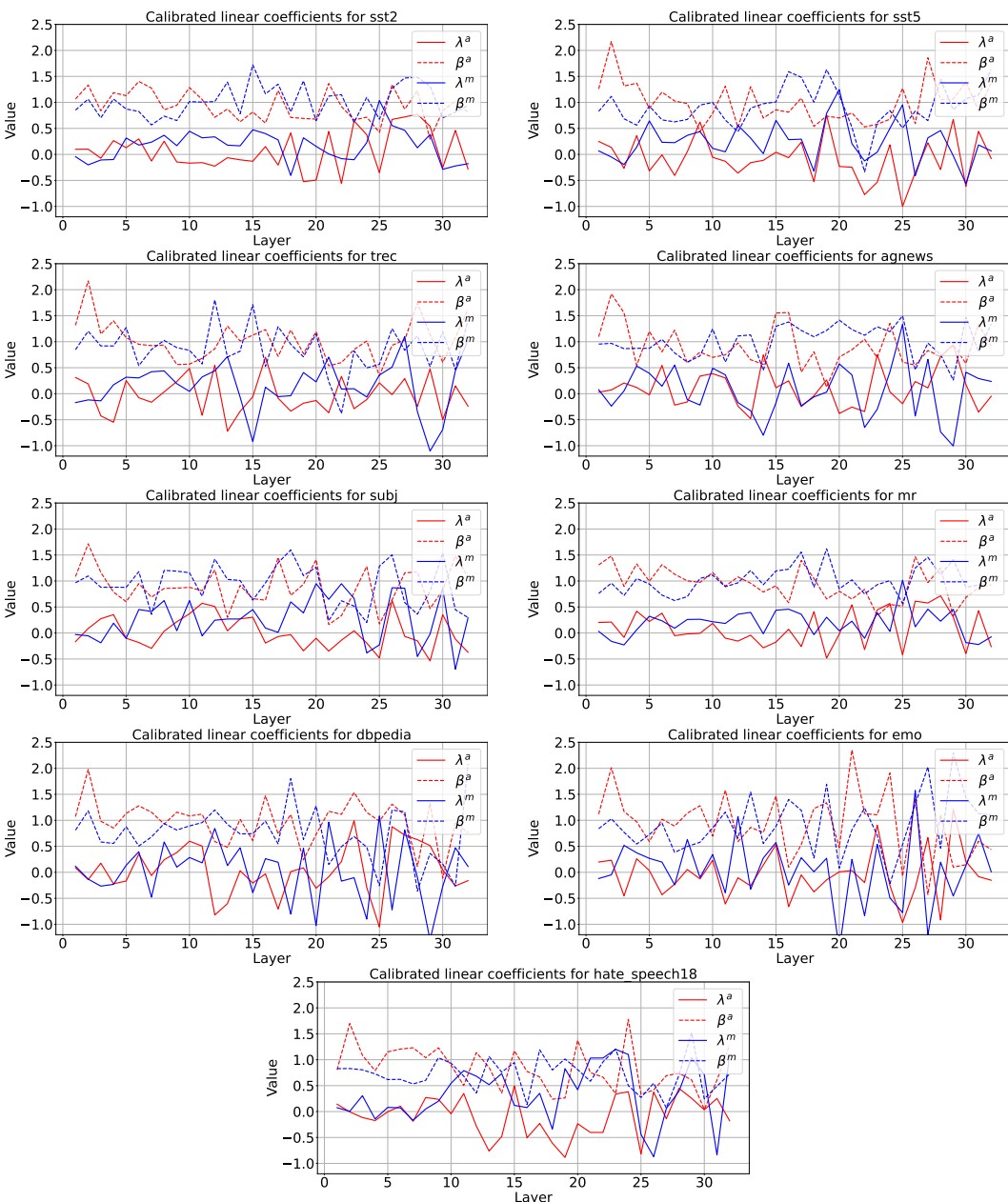

Figure 6: Calibrated coefficients for various datasets.

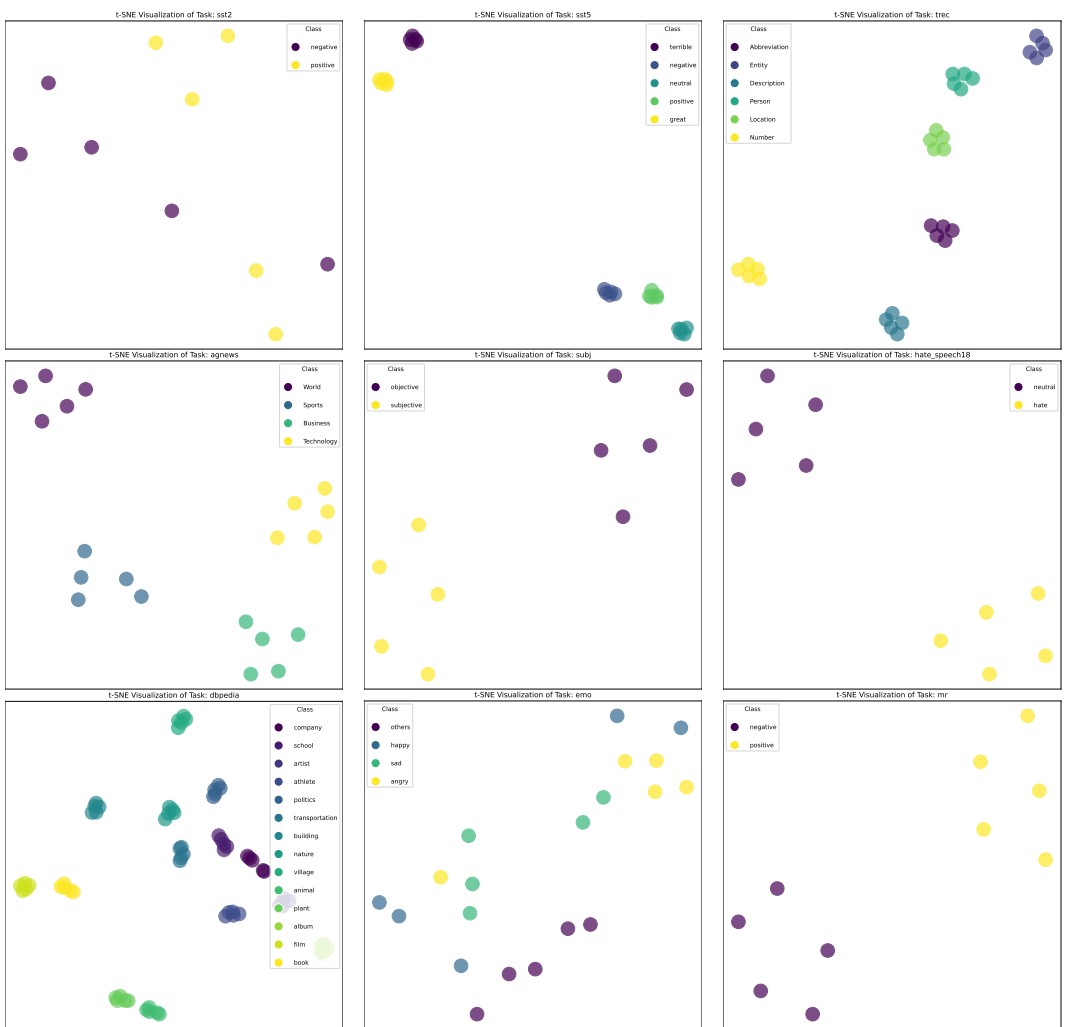

Figure 7: Visualization of in-task context vectors.

