# OpenReview forum: "Implicit In-context Learning"
_ICLR.cc/2025/Conference — ICLR 2025 Poster_

### Official Review · Reviewer_APKm · 2024-10-27

**Soundness:** 4
**Presentation:** 3
**Contribution:** 2
**Rating:** 6
**Confidence:** 3

**Summary:**

This paper proposes an innovative paradigm that reduces the inference cost of ICL to that of zero-shot learning with minimal information loss. The results reveal that the method is quite effective with certain extra benefits including enhancing task similarity detection and fostering effective transfer learning.

**Strengths:**

1. The proposed method is simple, interesting and effective.
2. Experimental results demonstrate the method's superiority.
3.  The paper is well-structured and easy to comprehend.

**Weaknesses:**

1.  The paper relies solely on experimental evidence to validate the superiority of the proposed method, lacking a clear explanation of why it outperforms prior approaches. Additionally, there is a lack of detailed analysis, making it feel more like a technical report.
2.  More ablation studies are needed to provide deeper insights.

**Questions:**

1. The proposed method performs worse than the prompt-compression-based method on some tasks but outperforms it on others. I'm curious about which task characteristics contribute to these differences in performance.

---

### Official Review · Reviewer_m8x3 · 2024-10-28

**Soundness:** 3
**Presentation:** 4
**Contribution:** 2
**Rating:** 6
**Confidence:** 3

**Summary:**

This paper introduces presents a method to reduce the computational and memory overheads of in-context learning. The proposed method, called implicit in-context learning (I2CL), extracts a context vector from demonstration examples and inject the context vector for the inference of query example. Experiments on 8 text-classification datasets show that I2CL successfully reduces the computation and memory cost of ICL, while performs only slightly worse than vallina ICL. The authors also show that I2CL is less sensitive to the order of demonstration examples.

**Strengths:**

1. This paper is clear and eazy to follow. The plot and table are eazy to understand
2. The proposed method is simple, and it achieves improvement on the inference speed and memory cost, compared to vallina ICL
3. The proposed algorithm is less sensitive to the order of demonstration examples.
4. The experiment part is comprehensive. The method is compared to various baselines. The authors also studied careful ablation studies to understand  the method.
5. The observation that the calibrated  linear coefficients can function as effective task-ids is interesing

**Weaknesses:**

1. The major claim of this paper is that I2CL achieves ICL performance at zero-shot cost in terms of both memory usage and inference speed, which raises my concern on the fairness of the comparison. Does the optimization of noisy self-calibration (Sec 2.4) included in the comparision? The calibration process requires inference of LLMs on each demonstration examples multiple times and optimize the linear coefficient with Adam. In comparison, ICL only requires forward pass of LLM on one-time. In table 1, only #cached param are considered as memory cost, and the cost of noisy calibration is missing. Also, is noisy self-calibration counted as inference time (the author metioned the optimization takes 1-2 minutes on a single A100?
2. In terms of the performance, I2CL is 1% worse than ICL on average. On simple task such as SST2 and MR, the performance of I2CL is not satisfying. Is there any specific reason that cause the degraded performance?
3. Only one very simple prompt is used in this paper. Have the author tried other prompts? From my own experience, SUBJ would be a very simple dataset for Llama-2 few-shot setting given some basic prompts.
4. On the scaling plot of figure 3: what dataset is the figure using? And as shown [1], the performance of ICL is expected to increasing as the number of demonstration increases. Could the author explain why the performance of ICL starts to decrease after 5-shots?

**Questions:**

Please refer to the weakness.

---

### Official Review · Reviewer_9isd · 2024-11-03

**Soundness:** 3
**Presentation:** 3
**Contribution:** 3
**Rating:** 8
**Confidence:** 4

**Summary:**

This paper introduces a mechanism (I2CL) to mitigate major criticism of in-context learning which is the inference cost associated with it as we need to pre-append all the demonstration examples always while performing inference. The proposed method involves two step first involving generating a context vector based on each demonstration by feeding it through the LLM and caching the activation across each layer and second where we linearly combine these activations with query activations during inference using pre-learned weights (or coefficients). These coefficients are learned only using the few-shot examples available to us for in-context learning. They show that this method can achieve comparable performance to in-context learning and at the cost of zero-shot learning. The paper also provides various ablations on how generalizable learned coefficients are to un-seen samples and the robustness of I2CL in comparison to in-context learning in different scenarios such as ordering of samples, choice of samples etc.

**Strengths:**

- Proposes a method for performing in-context learning at the cost of zero-shot learning and their method (I2CL) performs comparable to in-context learning.
- The idea of performing activation merging is an interesting direction to make model learn new skills at a very negligible cost.
- I2CL scales very well with adding more samples in-contrast to in-context learning.
- The paper provides comparison of their method with diverse methods such as task vector, label anchors etc to show improvements and at the same time provide ablation studies showing the robustness of learned coefficients and the robustness of I2CL to ordering and choice of samples.
- Paper is well written and easy to understand.

**Weaknesses:**

- Learning of weights to infuse demonstration samples might require some compute and hence comparing this method to few-shot parameter efficient fine-tuning method would also be helpful. As there are two computational cost associated here one where you extract out the context vector and other where you train the coefficients to find optimal coefficients to combine samples.
- Caching large context vector (for larger models) might not be feasible which might make it harder to apply this method to very large models.

**Questions:**

- Could you please include a comparison with PEFT-based few-shot fine-tuning strategies to highlight how your approach performs relative to these methods?

---

### Official Review · Reviewer_sfPw · 2024-11-04

**Soundness:** 3
**Presentation:** 3
**Contribution:** 3
**Rating:** 6
**Confidence:** 3

**Summary:**

The paper I2CL proposes a method to achieve few-shot performance at zero-shot inference costs by generating a condensed "context vector" from demonstration examples, which is then injected directly into the model’s residual streams. Many experiments demonstrate the method’s efficacy.
The paper is well-structured and with sufficient experiments.
Managing similar cost with few-shot but keeping comparative performance sounds novel for me.

**Strengths:**

1. I2CL presents a unique approach to reducing ICL’s computational load, which is an important advancement, especially for applications with resource constraints.

2. The concept of "task-ids" is a novel contribution that could facilitate transfer learning.

**Weaknesses:**

1. I2CL’s dependence on accessing intermediate activations may limit its use in closed or black-box models where such access is restricted. Addressing this limitation or proposing adaptations for such environments would broaden its utility.

2. Expand the evaluation to include more diverse tasks, such as open-ended generation or multi-step reasoning. Demonstrating I2CL’s efficacy in these tasks would make the paper’s findings more universally applicable.

**Questions:**

1. The experiments are mainly based on Llama-2. Have you explored the potential of I2CL in larger language models? It would be helpful to understand whether the efficiency gains scale up with model size.
2. Just a question (not require you perform experiments in the revision period) and may help clarify the potential of I2CL, which is how to convert your method to vision area. Current visual in-context learning [Zhang et al.] requires extra training process to select best in-context samples. Do you think if I2Cl has the potential to transfer to visual modality?
3. It looks like context vectors are well seperated as shown in figure 4. How might the task-id approach handle more nuanced variations within a single task type? i.e., does such context vector carry some information regarding classification labels?


Zhang, Y., Zhou, K. and Liu, Z., 2023. What makes good examples for visual in-context learning?. Advances in Neural Information Processing Systems, 36, pp.17773-17794.

---

### Meta-Review · Area_Chair_j2zu · 2024-12-22

**Metareview:**

This work proposes a new implicit in-context learning that extracts a context vector from demonstration examples and injects the context vector into the model’s residual stream to reduce the inference cost of ICL to that of zero-shot learning  All reviewers consistently recommended accepting this work. AC agrees that this work is interesting and deserves to be published on ICLR 2O25. The reviewers did raise some valuable concerns that should be addressed in the final camera-ready version of the paper. The authors are encouraged to make the necessary changes in the final version.

**Additional Comments On Reviewer Discussion:**

All reviewers agee to accept this work.

---

### Decision · Program_Chairs · 2025-01-22

Accept (Poster)